# Different Methods for Assessing Tooth Colour—In Vitro Study

**DOI:** 10.3390/biomimetics8050384

**Published:** 2023-08-23

**Authors:** Susana Dias, Joana Dias, Ruben Pereira, João Silveira, António Mata, Duarte Marques

**Affiliations:** 1Oral Biology and Biochemistry Research Group, Faculdade de Medicina Dentária, Universidade de Lisboa, 1950-044 Lisboa, Portugal; dias.susana@campus.ul.pt (S.D.); joananogueiradias@gmail.com (J.D.); ruben-pereira@campus.ul.pt (R.P.); silveira@campus.ul.pt (J.S.); antonio.mata@fmd.ulisboa.pt (A.M.); 2LIBPhys-FCT UID/FIS/04559/2013, Faculdade de Medicina Dentária, Universidade de Lisboa, 1950-044 Lisboa, Portugal; 3Postgraduate Programme in Prosthodontics, Faculdade de Medicina Dentária, Universidade de Lisboa, 1950-044 Lisboa, Portugal

**Keywords:** tooth colour, eLAB, SpectroShade, spectrophotometry, photography

## Abstract

Colour assessment using digital methods can yield varying results, and it is important for clinicians to recognize the potential variability intra and inter-device. This study aimed to compare the L*a*b* values of VITA Classical (VC) and VITA Toothguide 3D-MASTER (VM) guides using two methods, SpectroShade (SS) and eLAB. Thirty-four measurements per tab were performed by a single operator across three batches of each guide. Intraclass correlation coefficients (ICC) between batches were calculated. Values <0.5, 0.5–0.75, 0.75–0.9, and >0.90 were classified as poor, moderate, good, and excellent reliability, respectively. Results were reported as mean and standard deviation of the L*a*b* values and respective colour differences (ΔE_00_) for each tab and method. Statistical analyses were performed with an independent *t*-test, α = 0.05. ICC values between batches were excellent for all L*a*b*, except for a* component in eLAB. There were statistically significant differences between methods in most L*a*b* values. The intra-device mean ΔE_00_ was 0.5 ± 0.6 for VC, 0.5 ± 0.8 for VM in SS, 1.1 ± 0.8 for VC, 1.1 ± 0.9 for VM in eLAB. The mean ΔE_00_ inter-device was 4.9 ± 1.7 for VC, 5.0 ± 1.7 for VM. Both methods demonstrated good internal consistency, with high ICC values and low intra-device colour differences, but exhibited high variability between methods, higher for a* the component.

## 1. Introduction

Colour assessment is a critical aspect of biomimetics in dentistry, playing a crucial role in clinical applications, research, and patient satisfaction [1,2,3,4,5,6]. Accurate determination of tooth colour ensures the successful integration of dental restorations with natural dentition [5,6]. However, this process remains subjective due to the involvement of light sources, objects, and observers [7,8].

To provide a quantifiable and measurable approach, the Commission Internationale de l’Eclairage (CIE, International Commission of Illumination) has adopted the CIELAB colour space [8,9], which is widely used in dentistry [4,6,10].

The CIELAB system consists of three coordinates: L* (luminosity) [6], a* (red-green axis), and b* (yellow-blue axis) [10,11]. The difference between two points in this colour space, known as ΔE, can be measured to assess perceptible and/or acceptable colour variations. In dentistry, perceptibility refers to the perception of a colour difference between a tooth and an adjacent restoration, whereas acceptability refers to whether the difference is deemed acceptable or questionable [1,8,12].

Various colour-difference formulas have been utilized in dentistry, with the CIEDE2000 (ΔE_00_) formula being considered more reliable and widely accepted, and its perceptibility and acceptability thresholds are set at 0.8 and 1.8, respectively [1,9,10].

Tooth colour assessment involves the process of shade matching, where the dentist selects the appropriate shade of the dental material to match the patient’s natural teeth [4,5,6]. This can be challenging due to the complex and subtle colour variations of teeth, as well as external factors influencing tooth colour, such as food, beverages, tobacco, medications, or trauma [5,8,13]. However, advancements in technology and shade-matching systems enable dentists to achieve more accurate and predictable results [5,8].

Visual and digital assessments are commonly used in dentistry to determine tooth colour [8,14,15]. Visual analysis using shade guides is frequently employed due to its simplicity and low cost, despite subjectivity owing to inter-operator variability, ambient influence, and VITA shade influence [9,11]. The two most popular shade guides are the VITA Classical (VC) and the VITA Toothguide 3D-MASTER (VM) [6,10]. The VC shade guide consists of 16 tabs, grouped into A, B, C, and D, representing reddish, yellowish, reddish-grey, and yellowish-grey teeth, respectively. Each letter corresponds to a different hue, with each hue having a value assigned for increasing pigment saturation. However, the increments of colour gradients are arbitrary, making it difficult to accurately reproduce the shade code of the tab [2]. The VC shade guide has limitations, such as an inadequate range of shades and unsystematic colour differences [7]. On the other hand, the VM was designed to overcome these limitations by offering a broader and more uniform colour range, better colour distribution, and improved reproducibility when compared to other shade guides [6,7]. It consists of 29 tabs ordered according to the parameters of lightness, chroma, and hue, divided into five value groups ranging from 1 to 5 [11,16].

Due to the limitations of visual colour determination, instrumental methods, such as spectrophotometers, colourimeters, or digital photography, have gained popularity due to their higher accuracy [4,8,15]. However, these instrumental methods have a steep learning curve due to their complexity and require specific and expensive technology, which may not always be available to clinicians [8,17,18].

SpectroShade Micro (SS, MHT Medical High Technologies, Bologna, Italy), widely used, is an imaging dental spectrophotometer that combines a digital camera and LED spectrophotometer, offering better accuracy compared to devices like EasyShade (VITA Zahnfabrik, Bad Säckingen, Germany) [16,18].

Digital cameras have diverse applications in dentistry, including tooth colour quantification, colour communication improvement [6], and capturing tooth and surrounding tissue details [8,19]. However, calibration procedures and colour adjustments are necessary to enhance digital photography [20].

The eLAB system, based on CIELAB, is designed for tooth colour measurement using a standardized dental photography protocol [9]. This system employs cross-polarisation to eliminate brightness and specular reflections [9,11], a grey reference card, and standard camera settings [9]. The captured cross-polarized image can be calibrated using the eLAB_prime application [9]. The eLAB protocol complements clinician skills and guides ceramists toward clinically acceptable shade matches [9].

Nevertheless, factors like variations in shade guide tabs and unspecified L*a*b* values for software calibration may affect colour assessment. These objective methods may yield different results, highlighting the need for testing and comparing tooth colour assessment using digital methods to ensure consistent colour conveyance among clinicians and detect potential discrepancies [18,21].

This study aims to compare two digital methods, SpectroShade Micro and the eLAB photographic method, in assessing tooth colour using VITA Classical (VC) and VITA Toothguide 3D-MASTER (VM) shade guides. The L*a*b* values and ΔE_00_ values will be determined and compared between the two methods. The null hypothesis assumes no statistically significant differences in the results between the two digital methods for both shade guides. Additionally, the study will evaluate the variability among three different batches of each shade guide.

## 2. Materials and Methods

An in vitro protocol was established for tooth colour measurements using two distinct digital methods: spectrophotometry with the SpectroShade Micro (SN: HDL3973) and digital photography with computer software—eLAB system (eLABor_aid^®^ System, Emulation, Freiburg, Germany).

In this study, two shade guide systems were evaluated: the VITA Classical (VC, VITA Zahnfabrik, Bad Säckingen, Germany) shade guide, featuring 16 colour tabs, and the VITA Toothguide 3D-MASTER (VM, VITA Zahnfabrik, Bad Säckingen, Germany) shade guide, comprising 29 colour tabs (including VITA Bleached Shades). To account for potential manufacturing variations, three distinct batches of VC (B027C; B027CV1; B27C) and VM (B360ASP; B260ASP; B360APOR) were utilized.

A calibrated operator conducted 34 consecutive measurements of each tab from the three different batches of VC and VM using the two digital methods (SS and eLAB), resulting in a total of 102 measurements per tab. All measurements were performed in a dark chamber following a predefined methodology [17,18,21].

The digital systems were operated in accordance with established protocols and the manufacturer’s instructions. Before measuring each tab, the SS device was calibrated using white and green tiles, with the optical piece positioned at a 90° angle against the gingival matrix.

For the eLAB protocol, the photographs were taken using a Reflex Canon EOS 1300D camera, a 100 mm macro F2.8L lens, a Canon Macro Twin Lite MT-26EX-RT flash, a cross polarizer filter (Polar_eyes^®^), and an eLAB_prime white balance card (Emulation, Freiburg, Germany).

The images were captured and subsequently analysed following the manufacturer’s instructions. These images were then imported into Adobe Photoshop Lightroom^®^ software (6.0 macOS, Adobe Inc., San Jose, CA, USA) in RAW format. After importation, the appropriate DSLR camera profile from the Lens Corrections menu was selected in the Develop mode (Figure 1). The white balance tool (pipette) was used by clicking on any of the four grey segments in the images to perform white balance correction (Figure 2). Exposure balance was achieved by selecting the three zeros next to the exposure slider and adjusting image exposure using the up(pipette)/down arrow on the keyboard until the known luminosity value of the grey reference card (L*79) was replicated (Figure 3) [22].

To determine the L*, a*, and b* values for each photograph and tab, a pre-established grid was used, and four corresponding centre points were obtained. The measurement was carried out using the Classic Colour Meter^®^ software (2.1.1 macOS, Ricci Adams, Cupertino, CA, USA). The grid was generated in the Adobe Photoshop Lightroom^®^ software (6.0 macOS, Adobe Inc., San Jose, CA, USA) (View > Loupe Overlay > Grid > Size 40) (Figure 4).

The sample size for this study was determined based on the colour difference (ΔE_00_) recorded in a previous study [18] (https://sample-size.net/sample-size-means/) (accessed on 19 September 2021). The sample size calculation considered a ΔE_00_ difference of 0.5 with a standard deviation of 0.5 for both VC and VM shade guides [18]. To achieve a significance level of 5% and a power of 80%, a minimum of 34 measurements per tab in each shade guide would be required, considering a T statistic and non-centrality parameter. In this study, a total of 102 measurements were performed per tab, including three shade guides per group.

The agreement and reproducibility of different batches of each shade guide were evaluated using the Intraclass Correlation Coefficient (ICC) with a 95% confidence interval of (CI 95%). The interpretation of ICC values used in this study was as follows: excellent (>0.9), good (0.75–0.9), moderate (0.5–0.75), and poor (<0.5) reliability [23]. To account for the variability between batches of the same guide, data from all three batches of each shade guide were analysed together if the obtained ICC was higher than 80% [18].

The colour differences for each tab were determined by ΔE_00_ (intra-method, inter-method global, and for each component, ΔL*, Δa*, Δb*, calculated with the CIE/ISO new standard, CIEDE2000 formula from the Commission Internationale De l’Eclairage. Computation with this colour difference formula was performed according to the following Equation (1):(1)$$\%\;\Delta E_{00}=\sqrt{{\left(\frac{L_{2}-L_{1}}{K_{L}S_{L}}\right)}^{2}+{\left(\frac{C_{2}-C_{1}}{K_{C}S_{C}}\right)}^{2}+{\left(\frac{H_{2}-H_{1}}{K_{H}S_{H}}\right)}^{2}+R_{T}(\frac{C_{2}-C_{1}}{K_{C}S_{C}})(\frac{H_{2}-H_{1}}{K_{H}S_{H}})}$$

To assess the perception of colour difference, two major thresholds were used: the perceptibility threshold (PT), defined as ΔE_00_ = 0.8, and the acceptability threshold (AT), considered as ΔE_00_ = 1.8 [1,24].

The results were presented as the mean and standard deviation (σ) of Lab values for each shade tab, along with the ΔE_00_ values between the two methods. Data were inputted and analysed using the statistical software SPSS (IBM Statistics v.25, Inc., Chicago, IL, USA). Parametric tests were employed when the minimum sample size of 30 was achieved, according to the central limit theorem. Independent *t*-tests were conducted to analyse colour coordinate values and differences with a significance level of 0.05.

## 3. Results

The measurement data presented in this study involved a total of 1632 evaluations for VC and 2958 measurements for VM for each method.

The agreement between each component measurement (L*, a*, b*) and the shade guides VC and VM for each method was assessed and recorded in Table 1 (SS) and Table 2 (eLAB). The analysis of data from three different batches resulted in a combined total of 102 measurements for each tab, demonstrating a strong ICC agreement. The eLAB method showed the lowest ICC value of 86%, while the SS method displayed the highest ICC value of 97%, indicating excellent agreement [22].

The mean and standard deviation of L*a*b* and ΔE_00_ values, along with their respective statistical significances, are presented in Table 3 (VC) and Table 4 (VM). The total mean of ΔE_00_ for eLAB was 1.1 ± 0.8 for VC and 1.1 ± 0.9 for VM, while for SS, it was 0.5 ± 0.6 for VC and 0.5 ± 0.8 for VM. The inter-device ΔE_00_ was 4.9 ± 1.7 for VC and 5.0 ± 1.7 for VM, with statistical differences between the devices exceeding the AT for all shade tabs in both VC and VM. Statistically significant differences were observed between the two methods for all L*a*b* values, except for L* for D4, 2M2, 3R1.5, 3M2, 3R2.5, 4L1.5, and 4L2.5; a* for 2M2, and 3R1.5; and b* for D4, 3R1.5, 3M2, and 3R2.5.

## 4. Discussion

In this in vitro study, two distinct digital methods, the SS spectrophotometer and the eLAB photograph/software measuring system, were employed to evaluate the tooth colour of two shade guides. Results showed significant differences in L*a*b* values and corresponding ΔE_00_ between the tested methods for both shade guides, leading to the rejection of the null hypothesis.

Regarding global intra-method ΔE_00_ colour differences, SS exhibited lower internal variability than eLAB. While neither method exceeded the acceptability limit (ΔE_00_ = 1.8) for the shade guides, they did surpass the perceptibility threshold (ΔE_00_ = 0.8). Possible explanations for this divergence could include a more sensitive protocol in eLAB, calibration challenges, or a greater number of steps in the eLAB protocol. Nevertheless, these values were still lower than those obtained by visual methods, indicating enhanced consistency and reproducibility of colour evaluation [10,14].

The global ΔE_00_ between methods was nearly three times higher than the acceptability threshold, which would be considered unacceptable in a clinical setting. Furthermore, statistically significant differences were observed between the methods for all L*, a*, and b* components. These disparities may be due to variations in protocols, different L*, a*, and b* values used for software calibration, and inherent variability between the methods. These findings highlight a weak relationship between the L*, a*, and b* values of the two methods, advising against making direct correspondences between them.

The ICC exhibited good to excellent values in both methods, with the lowest values observed for the a* component in SS for VC and eLAB for VM. These discrepancies could be attributed to inconsistencies between tabs or intrinsic variability among the batches [25,26]. Factors such as surface texture, tab thickness, and substrate colour can affect how light interacts with the shade tab, leading to differences in colour measurements [5,15,20]. Nevertheless, the devices may have intrinsic variability that can influence colour assessment, such as variations in fabrication processes, calibration procedures, or maintenance issues [18].

Although the devices measured the same shade tabs, they exhibited chromatic differences. This divergence could be due to the different approaches used to acquire colour data. SS can acquire colour in three different ways, while the eLAB protocol used in this study focused only on measurements of the central region of each Table. The use of the tooth average colour in SS may have compromised accuracy in this in vitro study, as compared to the eLAB protocol. Similar studies have also found that spot measurement can be more reliable and repeatable than an average measurement. However, a low degree of compatibility with digital devices from different manufacturers is always reported [27,28]. This inter-device variability can lead to problems in communication with dental technicians, who may evaluate the colour of the prosthetic work using a device divergent from the one initially used by the dentist during the colour selection process [18,25,27,28]. Additionally, differences in L*, a*, and b* values between shade guides could contribute to the data variation. However, if the devices showed differences when measuring similar samples, one could anticipate an increase in measurement variability when assessing natural teeth with factors like enamel thickness, dentin shade, and the presence of stains or restorations [11,15].

Based on the findings of this study, it can be concluded that the correspondence of L*, a*, and b* values between different methods for colour communication is not accurate. To ensure more precise and reproducible colour communication in a clinical setting, manufacturers should aim to improve standardisation [18]. Although this study compared two different methods for colour assessment using standardised shade guides in a controlled environment, further in vivo studies are required to evaluate the impact of these methods in a clinical setting—particularly the eLAB protocol.

## 5. Conclusions

The two digital methods presented high ICC values and low intra-device colour differences, indicating strong internal consistency and the viability of these methods for colour assessment. However, a notable inter-device variability was observed, particularly with the a* component. Caution is recommended when attempting to match measurements from these devices. Further studies are essential to comprehend the clinical implications of this variability.

## Figures and Tables

**Figure 1 biomimetics-08-00384-f001:**
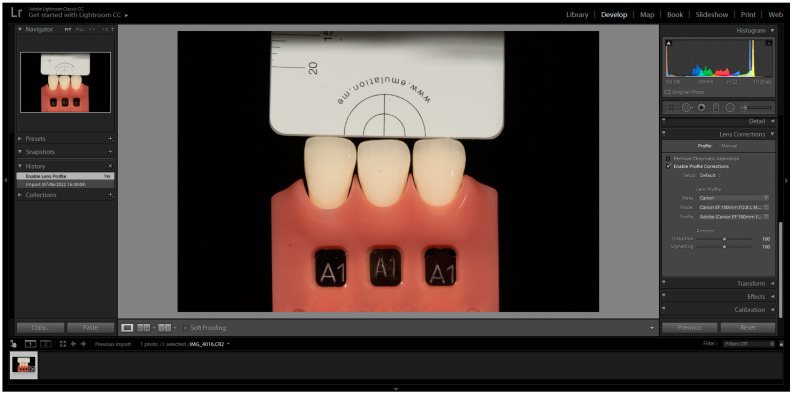
Selection of the correct DSLR camera profile from the camera calibration dropdown menu in the Develop mode.

**Figure 2 biomimetics-08-00384-f002:**
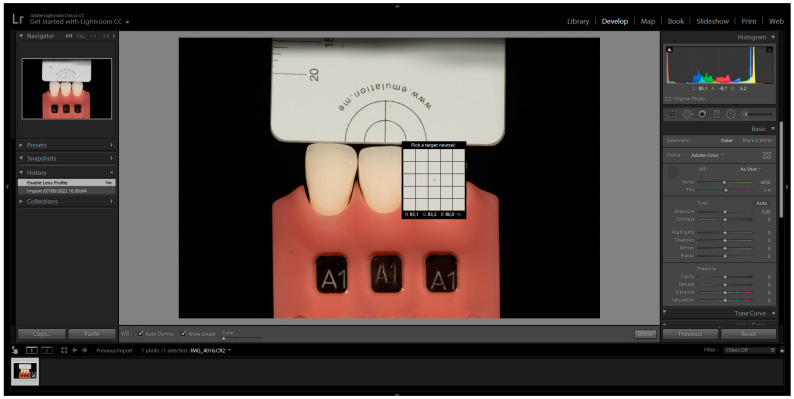
White Balance Correction with the selection of the pipette and clicking on any of the four gray segments.

**Figure 3 biomimetics-08-00384-f003:**
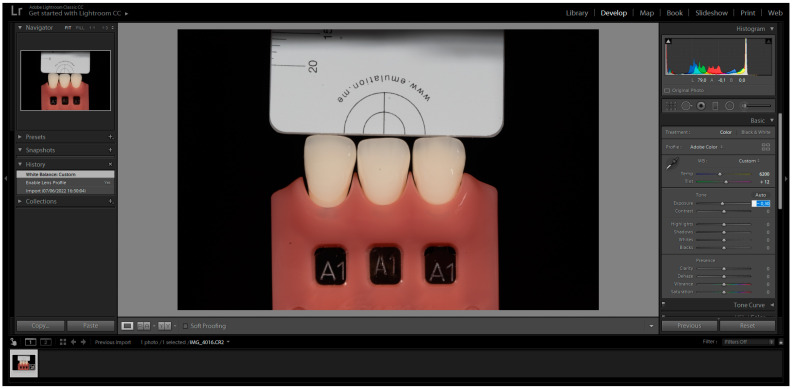
Exposure Correction clicking on the three zeros next to the Exposure slider. The cursor becomes a magnifying glass when moved over any of the four gray segments.

**Figure 4 biomimetics-08-00384-f004:**
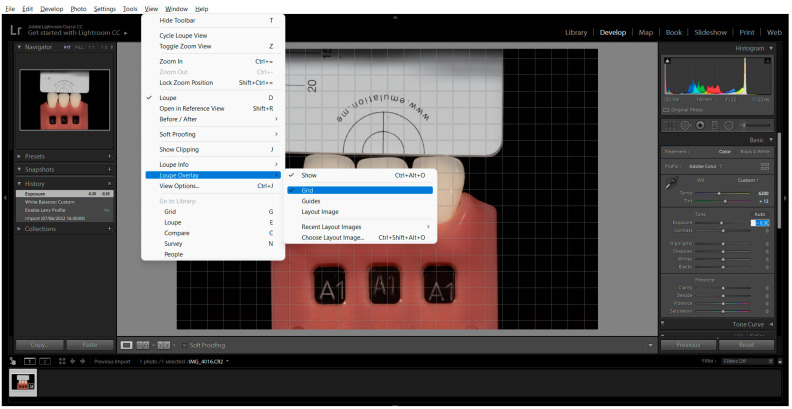
Measurement Grid placement by clicking on the Loupe Overlay menu and selecting Grid.

**Table 1 biomimetics-08-00384-t001:** Mean and confidence intervals 95% of Intraclass Correlation Coefficient (ICC) between the three batches for each component L*, a*, and b* of each shade guide, VITA Classical (VC) and VITA Toothguide 3D-MASTER (VM), for SpectroShade.

	VC		VM	
	ICC	CI 95%	ICC	CI 95%
L*	0.984	[0.982–0.987]	0.990	[0.989–0.991]
a*	0.972	[0.968–0.976]	0.991	[0.990–0.992]
b*	0.983	[0.980–0.985]	0.993	[0.992–0.993]

**Table 2 biomimetics-08-00384-t002:** Mean and confidence intervals 95% of Intraclass Correlation Coefficient (ICC) between the three batches for each component L*, a*, and b* of each shade guide, VITA Classical (VC), and VITA Toothguide 3D-MASTER (VM), for eLAB.

	VC		VM	
	ICC	CI 95%	ICC	CI 95%
L*	0.968	[0.964–0.973]	0.970	[0.967–0.973]
a*	0.970	[0.965–0.974]	0.862	[0.848–0.875]
b*	0.982	[0.979–0.984]	0.988	[0.987–0.990]

**Table 3 biomimetics-08-00384-t003:** eLAB and SpectroShade (SS) L*a*b* mean, and standard deviation (SD) values comparison, intra-device (ΔE_00_ eLAB and ΔE_00_ SS), and inter-device ΔE_00_ colour difference for each tab of VITA Classical (VC). Results in bold are statistically significant.

VC		SS	ΔE_00_ SS	eLAB	ΔE_00_ eLAB	*p*	ΔE_00_
B1	L*	75.7 ± 0.4	0.2 ± 0.5	74.5 ± 1.4	1.1 ± 1.3	**<0.05**	3.9 ± 0.5
	a*	−0.9 ± 0.1		1.9 ± 0.3			
	b*	12.8 ± 0.1		12.5 ± 0.5			
A1	L*	76.5 ± 0.3	0.2 ± 0.4	76.7 ± 2.3	1.6 ± 1.3	**<0.05**	3.7 ± 0.8
	a*	−0.3 ± 0.1		2.2 ± 0.3			
	b*	14.1 ± 0.2		12.9 ± 0.7			
B2	L*	74.2 ± 0.3	0.2 ± 0.4	72.7 ± 0.7	0.9 ± 0.8	**<0.05**	2.9 ± 0.5
	a*	0.2 ± 0.1		2.2 ± 0.3			
	b*	18.3 ± 0.2		17.4 ± 0.7			
D2	L*	69.7 ± 0.5	0.4 ± 0.5	67.2 ± 0.8	1.0 ± 0.5	**<0.05**	5.0 ± 0.7
	a*	0.7 ± 0.1		4.2 ± 0.5			
	b*	13.3 ± 0.2		13.9 ± 0.8			
A2	L*	73.8 ± 0.5	0.3 ± 0.4	72.6 ± 0.5	0.7 ± 0.6	**<0.05**	4.2 ± 0.4
	a*	1.1 ± 0.1		4.3 ± 0.3			
	b*	18.2 ± 0.2		18.6 ± 0.4			
C1	L*	70.8 ± 0.5	0.3 ± 0.5	68.9 ± 0.9	0.9 ± 0.5	**<0.05**	5.1 ± 0.6
	a*	0.0 ± 0.1		3.7 ± 0.4			
	b*	13.8 ± 0.2		15.2 ± 0.5			
C2	L*	69.0 ± 0.6	0.7 ± 0.5	66.4 ± 1.3	1.2 ± 0.8	**<0.05**	4.9 ± 0.9
	a*	1.1 ± 0.2		4.4 ± 0.5			
	b*	18.3 ± 0.8		19.9 ± 0.6			
D4	L*	66.9 ± 0.9	0.9 ± 0.4	66.2 ± 0.8	1.1 ± 1.0	0.750	3.4 ± 0.6
	a*	1.0 ± 0.3		3.6 ± 0.3		**<0.05**	
	b*	19.4 ± 0.7		21.1 ± 0.7		0.734	
A3	L*	71.7 ± 0.7	0.4 ± 0.8	68.6 ± 1.0	1.2 ± 0.7	**<0.05**	4.9 ± 0.6
	a*	1.8 ± 0.2		5.3 ± 0.5			
	b*	20.5 ± 0.4		21.5 ± 1.1			
D3	L*	68.9 ± 0.5	0.3 ± 0.5	68.3 ± 1.0	1.2 ± 1.1	**<0.05**	5.0 ± 0.7
	a*	1.4 ± 0.1		5.1 ± 0.3			
	b*	17.2 ± 0.1		18.9 ± 0.5			
B3	L*	71.0 ± 0.3	0.3 ± 0.3	69.7 ± 0.8	0.8 ± 0.5	**<0.05**	4.0 ± 0.5
	a*	1.6 ± 0.2		5.0 ± 0.4			
	b*	23.5 ± 0.2		25.2 ± 0.7			
A3.5	L*	68.9 ± 0.5	0.3 ± 0.6	66.0 ± 1.0	1.1 ± 0.7	**<0.05**	5.8 ± 0.7
	a*	2.4 ± 0.1		7.1 ± 0.7			
	b*	23.0 ± 0.3		26.3 ± 1.1			
B4	L*	69.7 ± 0.5	0.8 ± 0.4	66.8 ± 0.9	0.8 ± 0.4	**<0.05**	5.6 ± 0.7
	a*	2.1 ± 0.2		6.6 ± 0.4			
	b*	24.6 ± 1.1		28.2 ± 0.7			
C3	L*	66.4 ± 0.4	0.3 ± 0.5	62.8 ± 1.1	1.2 ± 0.6	**<0.05**	5.7 ± 1.1
	a*	1.2 ± 0.1		5.6 ± 0.7			
	b*	18.9 ± 0.1		21.2 ± 0.7			
A4	L*	65.4 ± 0.6	0.4 ± 0.6	61.6 ± 1.4	1.2 ± 0.8	**<0.05**	7.1 ± 0.9
	a*	3.2 ± 0.1		9.4 ± 0.5			
	b*	23.0 ± 0.1		26.7 ± 0.3			
C4	L*	61.2 ± 0.6	0.4 ± 0.6	57.0 ± 7.4	1.0 ± 0.5	**<0.05**	6.6 ± 1.2
	a*	2.5 ± 0.2		7.4 ± 0.6			
	b*	19.3 ± 0.3		22.5 ± 1.0			
Total			0.5 ± 0.6		1.1 ± 0.8		4.9 ± 1.7

**Table 4 biomimetics-08-00384-t004:** eLAB and SpectroShade (SS) L*a*b* mean, and standard deviation (SD) values comparison, intra-device (ΔE_00_ eLAB and ΔE_00_ SS), and inter-device ΔE_00_ colour difference for each tab of VITA Toothguide 3D-MASTER (VM). Results in bold are statistically significant.

VM		SS	ΔE_00_ SS	eLAB	ΔE_00_ eLAB	*p*	ΔE_00_
0M1	L*	81.1 ± 0.2	0.1 ± 0.2	78.0 ± 1.0	0.9 ± 0.6	**<0.05**	3.3 ± 0.5
	a*	−0.3 ± 0.1		0.8 ± 0.2			
	b*	6.7 ± 0.1		4.5 ± 0.3			
0M2	L*	79.8 ± 0.3	0.2 ± 0.2	7.7 ± 0.8	1.3 ± 2.6	**<0.05**	3.7 ± 3.1
	a*	−0.5 ± 0.1		1.6 ± 4.4			
	b*	8.0 ± 0.1		5.7 ± 0.4			
0M3	L*	79.2 ± 0.1	0.1 ± 0.2	77.2 ± 0.9	0.9 ± 0.7	**<0.05**	3.3 ± 0.6
	a*	−0.7 ± 0.0		0.8 ± 0.2			
	b*	9.4 ± 0.2		6.9 ± 0.4			
1M1	L*	77.5 ± 0.5	0.2 ± 0.6	74.7 ± 1.1	1.5 ± 2.7	**<0.05**	3.9 ± 3.1
	a*	−0.3 ± 0.0		1.8 ± 0.2			
	b*	12.3 ± 0.3		10.4 ± 0.3			
2M1	L*	72.5 ± 0.1	0.1 ± 0.2	71.0 ± 1.1	1.0 ± 0.7	**<0.05**	3.4 ± 0.5
	a*	0.2 ± 0.1		2.5 ± 0.2			
	b*	13.0 ± 0.1		12.5 ± 0.3			
1M2	L*	77.2 ± 0.4	0.2 ± 0.6	75.2 ± 1.5	1.2 ± 0.6	**<0.05**	3.5 ± 0.6
	a*	−0.5 ± 0.1		1.6 ± 0.2			
	b*	16.8 ± 0.2		15.4 ± 0.3			
2L1,5	L*	72.7 ± 0.0	0.3 ± 1.4	70.8 ± 0.8	0.9 ± 0.4	**<0.05**	4.1 ± 1.0
	a*	−0.5 ± 0.0		2.4 ± 0.5			
	b*	15.7 ± 1.4		16.1 ± 0.7			
2R1,5	L*	72.2 ± 0.3	0.3 ± 0.2	70.3 ± 1.2	1.1 ± 0.5	**<0.05**	3.7 ± 0.7
	a*	0.9 ± 0.1		3.4 ± 0.3			
	b*	15.1 ± 0.1		15.0 ± 0.5			
2M2	L*	71.8 ± 2.0	1.4 ± 1.4	72.0 ± 1,3	1.2 ± 0.6	0.053	4.0 ± 0.5
	a*	0.3 ± 0.4		3.1 ± 0.3		0.062	
	b*	17.0 ± 1.3		17.9 ± 0.6		**<0.05**	
3M1	L*	68.1 ± 0.2	0.2 ± 0.2	66.0 ± 1.3	1.2 ± 0.9	**<0.05**	4.2 ± 1.0
	a*	1.2 ± 0.1		4.1 ± 0.8			
	b*	14.2 ± 0.1		14.5 ± 0.5			
3L1,5	L*	67.0 ± 0.1	0.1 ± 0.3	65.4 ± 0.8	1.0 ± 0.6	**<0.05**	4.5 ± 0.6
	a*	1.1 ± 0.1		4.6 ± 0.5			
	b*	18.3 ± 0.2		19.6 ± 0.7			
2R2,5	L*	73.4 ± 0.4	0.2 ± 0.5	71.0 ± 1.0	0.9 ± 0.5	**<0.05**	4.4 ± 0.4
	a*	0.8 ± 0.0		3.9 ± 0.2			
	b*	20.9 ± 0.1		21.3 ± 0.3			
2L2,5	L*	72.3 ± 0.1	0.1 ± 0.5	71.3 ± 1.3	1.4 ± 0.6	**<0.05**	4.3 ± 0.9
	a*	−0.2 ± 0.1		3.1 ± 0.7			
	b*	22.4 ± 0.4		22.5 ± 1.5			
3R1,5	L*	71.8 ± 1.2	0.6 ± 1.4	66.4 ± 0.9	0.9 ± 0.7	0.448	7.3 ± 0.6
	a*	1.0 ± 0.3		5.5 ± 0.3		0.921	
	b*	15.2 ± 0.4		17.8 ± 0.4		0.161	
2M3	L*	73.2 ± 0.2	0.1 ± 0.4	71.8 ± 1.3	1.1 ± 0.6	**<0.05**	4.2 ± 0.6
	a*	0.3 ± 0.1		3.5 ± 0.4			
	b*	22.5 ± 0.1		23.4 ± 0.8			
3M2	L*	69.2 ± 1.6	1.0 ± 1.1	67.1 ± 1.3	1.0 ± 0.8	0.111	4.6 ± 0.5
	a*	1.7 ± 0.1		5.0 ± 0.3		**<0.05**	
	b*	20.1 ± 0.7		20.7 ± 0.6		0.652	
4M1	L*	65.2 ± 0.5	0.2 ± 0.6	61.0 ± 0.7	0.9 ± 0.9	**<0.05**	5.9 ± 0.4
	a*	1.9 ± 0.1		5.6 ± 0.2			
	b*	15.8 ± 0.1		15.9 ± 0.5			
3L2,5	L*	68.6 ± 0.3	0.2 ± 0.4	66.9 ± 0.8	0.9 ± 0.5	**<0.05**	4.8 ± 0.4
	a*	1.3 ± 0.1		5.2 ± 0.4			
	b*	23.3 ± 0.1		24.9 ± 0.6			
3R2,5	L*	70.0 ± 1.0	0.4 ± 1.3	61.1 ± 0.6	0.7 ± 0.5	0.082	5.5 ± 0.4
	a*	3.1 ± 0.2		6.7 ± 0.2		**<0.05**	
	b*	25.2 ± 0.8		24.8 ± 0.5		0.181	
4L1,5	L*	64.8 ± 0.5	0.2 ± 0.6	61.9 ± 0.4	0.8 ± 0.5	0.463	5.3 ± 0.5
	a*	2.2 ± 0.1		6.0 ± 0.4		**<0.05**	
	b*	20.1 ± 0.2		20.9 ± 0.8		**<0.05**	
3M3	L*	68.5 ± 0.1	0.3 ± 0.4	66.3 ± 1.1	1.2 ± 0.6	**<0.05**	4.5 ± 0.6
	a*	2.5 ± 0.1		6.0 ± 0.4			
	b*	26.0 ± 0.5		27.0 ± 1.0			
4R1,5	L*	65.0 ± 0.4	0.5 ± 0.5	61.9 ± 0.8	0.9 ± 0.5	**<0.05**	5.7 ± 0.9
	a*	3.1 ± 0.4		7.3 ± 0.5			
	b*	19.6 ± 0.3		19.7 ± 0.8			
4M2	L*	65.3 ± 0.4	0.5 ± 0.3	62.3 ± 0.6	0.9 ± 0.5	**<0.05**	6.2 ± 0.4
	a*	2.5 ± 0.3		7.3 ± 0.4			
	b*	20.7 ± 0.5		22.8 ± 1.0			
5M1	L*	60.1 ± 0.4	0.2 ± 0.5	55.6 ± 1.5	1.4 ± 0.8	**<0.05**	6.9 ± 0.8
	a*	3.2 ± 0.0		7.7 ± 0.4			
	b*	18.3 ± 0.2		18.6 ± 0.3			
4L2,5	L*	65.6 ± 0.8	0.4 ± 1.1	61.7 ± 0.9	1.1 ± 0.6	0.494	6.3 ± 0.8
	a*	3.5 ± 0.2		8.0 ± 0.8		**<0.05**	
	b*	27.8 ± 0.5		28.0 ± 1.1		**<0.05**	
4R2,5	L*	65.4 ± 0.5	0.8 ± 1.0	61.3 ± 0.9	0.9 ± 0.5	**<0.05**	6.9 ± 0.7
	a*	3.8 ± 0.1		9.0 ± 0.4			
	b*	24.5 ± 1.6		26.1 ± 0.8			
4M3	L*	66.0 ± 0.4	0.2 ± 0.6	62.2 ± 1.4	1.2 ± 0.7	**<0.05**	6.9 ± 0.8
	a*	3.5 ± 0.1		9.1 ± 0.3			
	b*	28.1 ± 0.1		30.3 ± 1.0			
5M2	L*	61.2 ± 0.2	0.1 ± 0.4	57.7 ± 1.2	1.1 ± 0.6	**<0.05**	6.9 ± 0.8
	a*	4.6 ± 0.1		10.1 ± 0.4			
	b*	25.1 ± 0.3		26.1 ± 0.5			
5M3	L*	62.0 ± 0.5	0.2 ± 0.7	57.9 ± 1.1	1.2 ± 0.7	**<0.05**	7.9 ± 0.9
	a*	5.8 ± 0.1		12.5 ± 0.6			
	b*	32.2 ± 0.4		33.5 ± 0.7			
Total			0.5 ± 0.8		1.1 ± 0.9		5.0 ± 1.7

## Data Availability

Original data are available on request from the corresponding author.

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
