# Peer review of "Different Methods for Assessing Tooth Colour—In Vitro Study"

_biomimetics, 2023, doi:10.3390/biomimetics8050384_

Round 1

Reviewer 1 Report

This is very useful study form the dental point of view. The study is almost perfectly reported.

I found mainly minor flaws:

1. Please write why this study is novel and important. Please add a paragraph before the aim of the study.

2. Authors should write in Introduction that different external factors can affect teeth color eg. Yılmaz N, Baygin O, Tüzüner T, Turgut SN, Erbek SM. Evaluation of the effect of pediatric drugs and an oral rinse on primary teeth discoloration. Dent Med Probl. 2022;59(2):225–231. doi:10.17219/dmp/133406

3. Please write in Materials and methods what kind of international standards (ISO or ADA) have been used for to determine the L*, a*, b* values?

4. Authors have to discuss the following latest study strongly related to the topic: Åšmielecka M, Dorocka-Bobkowska B. Comparison of two optical devices used for artificial tooth color selection. Dent Med Probl. 2022;59(2):249–253. doi:10.17219/dmp/141147

5. Authors have to add a legend of used abbreviations below each Table and Figure. In my opinion captions for the figures are very limited and not friendly for potential readers. Authors have to improve the captions for Figures.

The English language is fine.

Author Response

“This is very useful study form the dental point of view. The study is almost perfectly reported.

I found mainly minor flaws:”

Point 1: “1. Please write why this study is novel and important. Please add a paragraph before the aim of the study.”

Response 1: It was added in the document.

Point 2: “2. Authors should write in Introduction that different external factors can affect teeth color eg. Yılmaz N, Baygin O, Tüzüner T, Turgut SN, Erbek SM. Evaluation of the effect of pediatric drugs and an oral rinse on primary teeth discoloration. Dent Med Probl. 2022;59(2):225–231. doi:10.17219/dmp/133406”

Response 2: It was elaborated in the introduction.

Point 3: “3. Please write in Materials and methods what kind of international standards (ISO or ADA) have been used for to determine the L*, a*, b* values?”

Response 3: It was included in the document.

Point 4: “4. Authors have to discuss the following latest study strongly related to the topic: Åšmielecka M, Dorocka-Bobkowska B. Comparison of two optical devices used for artificial tooth color selection. Dent Med Probl. 2022;59(2):249–253. doi:10.17219/dmp/141147”

Response 4: It was discussed in the document.

Point 5: “5. Authors have to add a legend of used abbreviations below each Table and Figure. In my opinion captions for the figures are very limited and not friendly for potential readers. Authors have to improve the captions for Figures.”

Response 5: Thank you for your remarks. It was edited in the document to clarify the terms.

Reviewer 2 Report

Different methods for assessing tooth colour – in vitro study

While the work is overall carried out well and the review support the conclusion, there are several issues that need attention and upon addressing those issues the paper can be accepted

1-    Firstly, there are numerous typos (overtyping) throughout the manuscript, all requiring attention (the abstract has such errors).  There are several grammatical errors that needed to be corrected. I urge the authors to thoroughly go through the entire manuscript and check every line for spelling, grammar, or sentence construction-related errors as without these measures the account is unreadable. 

2-    Explain each word for first time in the beginning and used its abbreviation after that [eg: Classical (VC), Vitapan 3D Master (VM)]

3-    The whole text should be revised and rearranged to be more clear, in particular the introduction and the discussion should be Extensively edited for English language and style.

4-      Proper conclusion outcome of all items to be presented in the manuscript

Author Response

Different methods for assessing tooth colour – in vitro study

While the work is overall carried out well and the review support the conclusion, there are several issues that need attention and upon addressing those issues the paper can be accepted”

Point 1: “1-    Firstly, there are numerous typos (overtyping) throughout the manuscript, all requiring attention (the abstract has such errors).  There are several grammatical errors that needed to be corrected. I urge the authors to thoroughly go through the entire manuscript and check every line for spelling, grammar, or sentence construction-related errors as without these measures the account is unreadable.”

Response 1: It was corrected throughout the document.

Point 2: “2-    Explain each word for first time in the beginning and used its abbreviation after that [eg: Classical (VC), Vitapan 3D Master (VM)]”

Response 2: It was modified in the document.

Point 3: “3-    The whole text should be revised and rearranged to be more clear, in particular the introduction and the discussion should be Extensively edited for English language and style.”

Response 3: It was amended throughout the document.

Point 4: “4-      Proper conclusion outcome of all items to be presented in the manuscript”

Response 4: Thank you for your observations, this point was modified in the conclusion.

Round 2

Reviewer 1 Report

The manuscript has been correctly revised. I don't have further comments.

The language required only minor revision.

Reviewer 2 Report

Dear editor,

The authors reply to all comments.

Best regards

Sanaa